# Therapeutic Challenges for Gastric Neuroendocrine Neoplasms: Take It or Leave It?

**DOI:** 10.3390/medicina59101757

**Published:** 2023-10-01

**Authors:** Federica Cavalcoli, Camilla Gallo, Lorenzo Andrea Coltro, Emanuele Rausa, Paolo Cantù, Pietro Invernizzi, Sara Massironi

**Affiliations:** 1Gastroenterology and Digestive Endoscopy Unit, Fondazione IRCCS Istituto Nazionale dei Tumori, 20133 Milan, Italy; federica.cavalcoli@istitutotumori.mi.it (F.C.); paolo.cantu@istitutotumori.mi.it (P.C.); 2Division of Gastroenterology, Fondazione IRCCS San Gerardo dei Tintori, University of Milano-Bicocca School of Medicine, 20900 Monza, Italy; c.gallo19@campus.unimib.it (C.G.);; 3Unit of Hereditary Digestive Tract Tumours, Department of Surgery, Fondazione IRCCS Istituto Nazionale dei Tumori, 20133 Milan, Italy; emanuele.rausa@istitutotumori.mi.it; 4Division of Gastroenterology, Center for Autoimmune Liver Diseases, Department of Medicine and Surgery, University of Milano-Bicocca, 20900 Monza, Italy; 5European Reference Network on Hepatological Diseases (ERN RARE-LIVER), San Gerardo Hospital, ASST Monza, 20900 Monza, Italy

**Keywords:** gastric neuroendocrine neoplasm, chronic atrophic gastritis, endoscopic resection, endoscopic mucosal resection, endoscopic submucosal dissection, systemic therapies

## Abstract

*Background and Objectives*: Gastric neuroendocrine neoplasms (gNENs) represent rare but increasingly recognized tumors. They are distinguished into three main clinical types (type-1, type-2, and type-3) according to gastrin level and at histological evaluation in well-differentiated G1, G2, or G3 lesions, as well as poorly-differentiated lesions. Small type-1 and type-2 neoplasms with low proliferation indices demonstrated excellent survival without progression during an extended follow-up period, and for these reasons, active endoscopic observation or endoscopic resection are feasible options. On the other hand, surgery is the treatment of choice for more aggressive type-3, G3, or infiltrating neoplasms. The present study aims to comprehensively review and compare the available therapeutic strategies for gNENs. *Materials and Methods*: A computerized literature search was performed using relevant keywords to identify all of the pertinent articles with particular attention to gNEN endoscopic treatment. *Results*: In recent years, different endoscopic resective techniques (such as endoscopic mucosal dissection, modified endoscopic mucosal resection, and endoscopic full-thickness resection) have been developed, showing a high rate of complete resection for advanced and more aggressive lesions. *Conclusions*: Overall, gNENs represent a heterogeneous group of lesions with varying behavior which require personalized management. The non-operative approach for small type-1 gNENs seems to be feasible and should be promoted. A step-up approach with minimally invasive endoscopic therapies might be proposed, particularly for type-1 gNEN. On the other hand, it is important to recognize the negative prognostic factors in order to identify those rare cases requiring more aggressive approaches. A possible therapeutic algorithm for localized gNEN management is provided.

## 1. Introduction

Gastric neuroendocrine neoplasms (gNENs) are relatively rare tumors, accounting for approximately 7% of all gastro-entero-pancreatic neuroendocrine neoplasms (GEP-NENs) and comprising less than 1% of all gastric malignancies [1]. In recent years, the diagnosis of gNENs has been increasing due to the widespread use of upper gastrointestinal endoscopy, increased disease awareness, and improved diagnostic techniques [1,2]. Currently, the age-adjusted incidence rate is about 0.2 cases per 100,000 individuals per year [3], with a mean age at presentation of 60–67 years [4,5]. GNENs are categorized into three main types (type-1, type-2, and type-3) with distinct etiologies, biological behaviors, and prognoses. Like other neuroendocrine neoplasms, they are classified as well-differentiated or poorly-differentiated neoplasms (also known as neuroendocrine carcinomas or NECs). Well-differentiated tumors can be further classified as G1, G2, or G3 neoplasms based on the Ki67 proliferation index and mitotic count (Ki67 < 3% or mitotic count <2 in 2 mm^2^, Ki67 3–20% or mitotic count 2–20 in 2 mm^2^, and Ki67 > 20% or mitotic count >20 in 2 mm^2^, respectively) [6,7,8].

Type-1 neoplasms constitute the majority of gNEN cases, accounting for almost 70–80% of instances [9]. These tumors arise from the abnormal proliferation of enterochromaffin-like (ECL) cells, driven by elevated circulating gastrin levels. Hypergastrinemia typically occurs in patients with chronic atrophic autoimmune gastritis (AIG), as a consequence of intragastric hypo-achlorhydria [10,11,12]. Consequently, type-1 gNENs represent a relevant complication of AIG, with an annual cumulative risk of approximately 5.7% according to recent data [13]. These neoplasms are generally small polypoid lesions, often multiple (in 65% of cases), primarily situated in the gastric body or fundus and mainly confined to the mucosa or submucosa [14]. Type-1 gNENs are typically well-differentiated, characterized by a Ki-67 < 1%, and present a metastasis risk of less than 5% [6,7]. Figure 1 rappresent a small type-1 gNEN.

Type-2 gNENs represent 5–6% of all gastric NENs. Similarly to type-1 neoplasms, they stem from abnormal ECL cell proliferation in response to the trophic effects of hypergastrinemia on the gastric mucosa. However, in this scenario, the hypergastrinemia results from excessive primary gastrin secretion due to a gastrinoma, leading to Zollinger–Ellison Syndrome (ZES) [15,16] and often occurring in patients with MEN1 syndrome [17,18]. Type-2 gNENs also tend to be small, multiple, and polypoid [14]. While they are generally considered relatively benign tumors, up to 30% of cases are diagnosed at a metastatic stage [6,7,19].

Type-3 gNENs constitute approximately 14–25% of all gNENs. These sporadic neoplasms are unrelated to hypergastrinemia and are not linked to enterochromaffin-like cell hyperplasia, gastric atrophy, or gastrinoma [20]. They are usually larger, solitary lesions, often poorly differentiated, and feature a high Ki-67 index. A type-3 gNEN is depicited in Figure 2. Deep wall invasion, microvascular and/or lymphatic invasion, and metastases may already be present at the time of diagnosis (in 50–100% of cases) [7,21].

It is worth noting that most gNENs are represented by type 1. These cases are characterized by benign behavior and the absence of distant metastases. Accordingly, therapeutic interventions for such cases are mainly focused on endoscopic strategies. In recent years, endoscopic procedures have been increasingly improved thanks to recent advances in the field, making them more and more suitable for achieving curative outcomes.

However, the spectrum of therapeutic approaches includes active endoscopic surveillance, endoscopic resection, surgery, pharmacological agents (such as somatostatin analogs, target therapy, and chemotherapy), and peptide receptor radionuclide therapy (PRRT). The choice of treatment hinges on various factors, including the patient’s clinical profile; characteristics of the tumor as observed through endoscopy, radiology, and histology; and the physician’s expertise. In situations where rigorous clinical trials are lacking, clinicians often rely on established clinical best practices.

The present study aims to comprehensively review and compare the available management and therapeutic strategies for the treatment of gNENs, and to provide a possible therapeutic algorithm for gNENs.

## 2. Materials and Methods

A computerized literature search was conducted in PubMed, Embase, SCOPUS, and Web of Science using a combination of both free language words and medical subject heading terms. The search terms included: gastric carcinoids type 1; gastric neuroendocrine tumors; gastric neuroendocrine neoplasm; chronic atrophic gastritis; endoscopic resection; endoscopic mucosal resection, endoscopic submucosal dissection, endoscopic full-thickness resection, treatment; therapy; and follow-up. The search strategy was last updated in March 2023. For “disease condition”, the following terms were used: (gastric) AND (neuroendocrine OR endocrine) AND (tumor OR tumour OR neoplasm) OR (carcinoid). The search also included the following terms for “treatment”: endoscopic resection OR endoscopic mucosal resection OR endoscopic submucosal dissection OR endoscopic full-thickness resection. The reference lists from the studies selected by the electronic search were then manually searched to identify further relevant reports. All the available primary studies, review articles, abstracts, and proceedings of relevant meetings were considered, whereas non-English language papers were excluded. The studies considered to be potentially eligible were retrieved as full texts and evaluated. In the case of duplicate publications, the most up-to-date versions were considered. Of all the selected articles, 41 were included in this review.

## 3. Therapeutic Approaches

### 3.1. Active Endoscopic Observation

Recent data highlight an increase in the incidence of neuroendocrine neoplasms, with a marked rise in their prevalence, although overall patient survival remains unaffected [1,22]. These trends suggest that the increased incidence of gastric neuroendocrine neoplasms is predominantly due to improved diagnostic sensitivity. The proportion of patients with metastatic disease remains relatively constant [1]. Notably, a study conducted across two referral centers for neuroendocrine neoplasms demonstrated that most (>90%) type-1 gNENs with a size of <10 mm do not progress towards aggressive forms during extended endoscopic follow-up [2]. Hence, an active endoscopic observation approach might be plausible for small, well-differentiated neoplasms with low proliferation Ki-67 indices. In line with the latest ENETS guidelines [8], type-1 gNENs under 10 mm G1 can be actively monitored without stringent biopsy repetition, unless unusual features emerge (e.g., ulceration, erosion, pitting).

The treatment of type-2 gNENs hinges on the management of Zollinger–Ellison Syndrome (ZES) and, if applicable, MEN-1 syndrome [15,16,23]. In cases of resectable gastrinoma and the presence of small G1 type-2 gNENs, a conservative approach with active observation, as recommended for type-1 neoplasms by ENETS, can be applied [24].

G2 gNENs with low Ki67 deserve a note. Although some studies propose that low-G2 gNENs with Ki67 < 10% behave comparably to G1 gNENs [25], a consensus regarding their treatment remains elusive due to limited evidence for this specific subclass. While it is plausible to consider that G2 gNENs with Ki67 < 10%, like G1 counterparts, could be subject to active endoscopic surveillance, these hypotheses remain speculative, lacking substantial scientific support; thus, further research is warranted.

In addition, the precise timing for endoscopic observation remains uncertain, and the European guidelines involve an initial follow-up at 6 months followed by subsequent annual assessments [8,26].

### 3.2. Endoscopic Resection

GNENs approaching 10 mm and/or those beyond G1 should be considered for resection. Larger lesions and/or those with higher Ki67 levels have shown increased local and distant metastatic risk, except for low-G2 gNENs [27]. In the specific case of type-2 gNENs, as stated above, the most important treatment is the resection of underlying gastrinoma, even if the gNEN may be endoscopically treated before the surgical resection [24].

Before resection, both type-1 and type-2 gNENs, if they exceed 10–15 mm or non-G1, should undergo endoscopic ultrasonography (EUS) [8] to assess local invasion depth and lymph nodes and to confirm their suitability for resection [28]. Endoscopic resection is considered appropriate when no lymph node metastases are detected at EUS, despite the latest European guidelines suggesting that muscularis propria invasion represents a contraindication for endoscopic resection [8]. These indications must, however, take into account the evidence that EUS presents low accuracy in the staging of submucosal lesions (45%) when compared with histological examination of the resected specimen [29].

In cases of type 3 gNEN, unfavorable locoregional behavior (such as submucosal and/or lymphovascular infiltration), larger lesions, and/or a higher Ki67 proliferation index, both EUS and further anatomical and functional imaging techniques like MRI, CT-scan, 68GalliumPET-CT, and FDG PET-CT should be used to assess distant metastasis, even if the optimal cutoff values for Ki-67 or size to perform a complete staging have not been identified [8,26].

Different endoscopic techniques for gNEN treatment have been described, including excisional biopsy, cold-snare or hot-snare polypectomy, traditional or modified endoscopic mucosal resection (EMR), endoscopic submucosal dissection (ESD), and endoscopic full-thickness resection (EFTR) [13]. The current data on endoscopic treatment are primarily based on type-1 gNENs. Traditional approaches like polypectomy and EMR are safe and feasible, while advanced methods like modified EMR, ESD, and EFTR exhibit higher R0 resection rates and lower recurrence risk. However, these advanced techniques are technically demanding, necessitate extended procedural time, and entail increased complication risk. The ideal indications for different endoscopic treatments remain to be determined, guided by lesion size, position, tumor invasion depth, and local endoscopist expertise. Regardless of the technique, achieving R0 resection is the overarching objective for all endoscopic procedures [30].

#### 3.2.1. Excisional Biopsy and Polypectomy

GNENs are frequently detected at very small sizes (<5 mm) [31]. For this reason, biopsy sampling, which serves diagnostic and staging purposes, can lead to complete resection [32]. While excisional biopsy shows no severe procedure-related complications [33], it is associated with a high recurrence rate (61.6%) over extended follow-up periods [34].

Cold snare polypectomy is a simple procedure in which the lesion is resected with a single-layer snare; it usually also provides the excision of 1–2 mm of normal mucosa around the lesion, and the margins of the resulting specimen are not altered by coagulation artifacts. Hot snare polypectomy, instead, involves the additional cutting power of electrocoagulation [35]. Data regarding snare polypectomy in gNENs are limited. In 2012, Merola et al. reported polypectomy use in 15 gNENs with complete resection in all cases; however, a high rate of recurrence, up to 63.6%, was observed [34]. This recurrence risk is partly due to the frequent involvement of the microscopic submucosal layer in this setting, which limits the effectiveness of these procedures [36]. Despite being technically feasible and safe for small gNENs, excisional biopsy and snare polypectomy are associated with elevated recurrence rates [37].

#### 3.2.2. Endoscopic Mucosal Resection

Endoscopic mucosal resection (EMR) is a minimally invasive procedure for the excision of luminal lesions; in the case of the conventional technique, it implies a proper lifting of the lesion followed by a hot snare resection of the lesion itself with electrocautery. The lifting of the lesion can be obtained by injecting saline solution; glycerol; or, in fewer cases, adrenalin in the submucosal layer, and the lesion may be excised en bloc or piecemeal [38]. Overall, EMR is a safe, cost-effective, and technically simple procedure; the median duration time of the procedure has been reported to range from 7.8 to 10 [39]. However, EMR has limits on achieving R0 resection for lesions that have submucosal involvement, such as gNENs [31]. A recent systemic review reported a rate of complete gNEN resection of 92.3% for conventional EMR, with R0 resection after en bloc removal of the lesion in 96.3% of cases [30]. This study reported an overall rate of complications of 5.4% for EMR [30]. The recurrence rate for EMR was reported to be 18.2% in a single study on type I gNEN after a median follow-up of 7 years [40], while Sivandzadeh et al. observed no cases of recurrence in a study that included 14 gNENs between 10 and 20 mm after 5 years of follow-up [41].

To address submucosal-involved lesions, technical EMR variants have emerged. Cap-assisted EMR (cEMR) is a device-assisted endoscopic procedure that allows for submucosal layer resection by applying a suction force directly to the lesion. To date, cEMR is usually performed using multi-band mucosectomy devices. In these procedures, the target lesion is suctioned into a transparent cap, then a rubber band is released by a controller, resulting in the creation of a pseudopolyp. The contraction of the rubber band at the base of the pseudopolyp is adequate to withhold the mucosa, but not the muscularis propria; thus, the injection of the submucosal space is not required. Finally, a snare is passed through the accessory channel and closed at the base of the pseudopolyp beneath the band; then, the pseudopolyp is resected using electrocautery [42]. The most used band ligation device is the Duette^®^ Multi-Band Mucosectomy device (Cook Medical, Limerick, Ireland) [43]. Recently, a new band ligation device has been launched (Captivator, Boston Scientific Ltd., Hemel Hempstead, UK), which would potentially allow for a better visualization through the cap and easier passage of accessories through the scope [44]. CEMR allows for the en bloc resection of small lesions up to 15 mm in diameter. For larger lesions, this system permits only piece-meal resections, which limits the pathologist’s ability to evaluate the lateral margins [42]. According to a recent study comparing different techniques for endoscopic resection in gastric, duodenal, and rectal NENs, cEMR with band ligation was demonstrated to have a higher en bloc resection rate when compared to conventional EMR (100% vs. 97%); adverse events were observed in 3.1% of cases, and these were represented by bleeding cases. Interestingly, cEMR without a band ligation device led to adverse events in up to 32% of the cases, suggesting that this procedure should be avoided [45].

Underwater EMR (uEMR) is performed by filling the gastric lumen with water and thus taking advantage of the ability of the water to lift the lesion [46]. Recently, Kim et al., in a small study on subepithelial lesions including gastric and duodenal NENs, reported en bloc resection rates and complete resection rates of around 100%. The median size of the resected lesions was 9 mm, and the mean procedural time was 3.2 min. No adverse events were observed, and no recurrence was found at a follow-up endoscopy after 3 months [47].

#### 3.2.3. Endoscopic Submucosal Dissection

Endoscopic submucosal dissection (ESD) is an advanced technique for the endoscopic resection of superficial gastrointestinal neoplasms. It is performed by delineating a circumferential excision zone around the lesion using an electrocauterization knife; then, fluid is injected into the submucosa to separate the lesion from the muscle layer. Finally, a dissection underneath the submucosal layer under direct visualization is performed [48]. Endoscopic submucosal dissection aims to achieve en bloc resection of the lesion through the inclusion of the submucosal layer underneath the lesion, thus increasing the chance of histologically complete resection [48]. The major advantages of ESD are the availability of submucosal tissue for pathological examination and the ability to perform en bloc resection even in cases of large lesions, of neoplasms with submucosal fibrosis, or of ulcerative non-lifting lesions [49]. In addition, examination of the submucosal tissue allows for the accurate determination of lymphatic invasion and histologic grading, which may guide subsequent therapeutic decisions [50]. However, ESD is a technically demanding procedure, and it is associated with a longer procedure duration and a higher risk of complications when compared with conventional endoscopic techniques [40]. In 2012, Chen et al. [50] reported the role of ESD in the management of 33 gNENs, including 22 type-1 and 11 type-3 lesions. R0 resection was achieved in 100% of cases, with horizontal and vertical negative margins and no lymphovascular invasions. The single adverse event which was observed was represented by delayed bleeding, which was successfully controlled endoscopically. During a median follow-up of 28.9 months, two patients presented local recurrences, and for both cases, the repetition of ESD was feasible. Accordingly, another recent study which analyzed the prognostic risk factors for ESD in 81 included gNENs showed a rate of complete resection of 100%, with an R0 resection rate of 88.9%. The mean procedure time was 18 min (range of 6–66). Adverse events were reported in 10.3% of cases, and, interestingly, the median procedure time proved to be related to post-operative bleeding [51]. Furthermore, according to a recent systematic review analyzing pooled data on endoscopic resection techniques for type-1 gNENs [30], R0 resection rates accounted for up to 97.4% of the included ESD cases, and en bloc resection was guaranteed in 98.7% of them. Adverse events were observed in 11.7% of cases, and, similarly to what was exposed in other studies, they were mostly represented by bleeding. The reported post-ESD rate of lesion recurrence was 11.5%; however, it must be considered that the median follow-up time was shorter than 30 months. With regard to the comparison of safety, feasibility, and radicality between the ESD and EMR procedures, a study by Kim et al., which included 87 cases of <10 mm type-1 gNENs undergoing endoscopic resection [47], reported that the R0 resection rates were 94.9% and 83.3% for ESD and EMR, respectively, with a rate of vertical margin involvement that was significantly lower in patients who underwent ESD (2.6% vs. 16.7%; *p* = 0.038). The adverse events rate was similar between the two groups (23.1% in the ESD group), and the events were mostly represented by bleeding; the only case of perforation was registered in the ESD group. All complications were successfully managed endoscopically [47]. Finally, with reference, again, to the systematic review conducted by Panzuto et al., no clear superiority of ESD over EMR in terms of efficacy and safety was reported, with similar R0 resection rates (97.4% and 92.3%, respectively) and complication rates (11.7% and 5.4%, respectively). Nevertheless, ESD demonstrated a lower risk of recurrence (18.2% vs. 11.5%, respectively).

#### 3.2.4. Endoscopic Full-Thickness Resection

Endoscopic full-thickness resection (EFTR) is an innovative endoscopic technique, originally proposed for the treatment of colonic lesions for which conventional endoscopic resection is not feasible. It is performed with the application of an over-the-scope clip (OVESCO^®^, Tübingen, Germany), and it allows for gut wall closure and resection of the tissue in one step, preventing perforation [52]. In recent years, the application of EFTR to the resection of non-colonic lesions, or the closure of wall defects, proved this technique to be feasible, effective, and safe. These results were also confirmed for gastric subepithelial tumors, but to date, data are on gNENs are very scarce [53]. To the best of our knowledge, the RESET trial reported the largest series of EFTR on gNENs. In this study on subepithelial gastric lesions, three gNENs of less than 15 mm were resected with the EFTR device, and R0 resection was obtained in all cases without major complications. No recurrence was detected at follow-up 3 months later [54]. Overall, very few cases of EFTR for gNEN have been reported to date; all reported cases showed high R0 resection rates and low rates of adverse events. Thus, EFTR seems to represent a promising technique for the treatment of gNENs; nevertheless, its indications still need to be clarified. Overall, endoscopic resection by ESD or EFTR has been proven to achieve higher R0 resection rates compared to EMR, but no randomized trials have compared the techniques head-to-head [8].

#### 3.2.5. Endoscopic Resection in Type-3 gNENs

Surgery is traditionally recommended for type-3 gNENs due to metastatic risk, although endoscopic resection’s role is emerging. Recent studies suggest that endoscopic resection is feasible for type-3 gNENs < 10–15 mm in size with low Ki67, demonstrating favorable outcomes. In 2013, Kwon et al. [55] retrospectively collected data from 50 patients with relatively small type-3 gNENs (<20 mm in size), treated endoscopically, with EMR or ESD in 41 and 9 patients, respectively. R0 resection was reported in 80.4% of cases (66.7% for ESD and 85.4% for EMR). Unfortunately, no data on Ki67% or mitotic count were reported. Three patients had lymphovascular invasion and, therefore, a surgical radicalization of the resection was performed. In the remaining patients, no local or distant recurrence was observed during the median 46 months follow-up, even in the case of incomplete resection. The authors thus suggested that endoscopic resection should be considered as a therapeutic option also for type-3 gNENs, especially if smaller than 20 mm, confined to the submucosal layer, and with a low Ki67 proliferation index. On the other hand, according to a paper by Min et al. [56], for well-differentiated G2 and G3 type-3 gNENs, limited to the submucosal layer, a poor prognosis after endoscopic resection was reported, with 42.9% of the patients having nodal or liver metastasis at a median of 59 months of follow-up. For these reasons, a more aggressive approach has traditionally been proposed, suggesting that only type-3 G1 gNENs no larger than 15 mm and limited to the submucosal layer should be considered for endoscopic resection (EMR or ESD) in the absence of lymphovascular invasion [56]. More recently, a Japanese multicenter retrospective study analyzed data from 144 patients with type-3 gNENs who underwent either surgical or endoscopic resection. Overall, 63 patients underwent endoscopic resection (53 ESD and 10 EMR); of these, 15 patients required additional surgery because of lymphovascular invasion, a positive vertical margin, and/or G2 histology. Interestingly, of the patients treated with endoscopic resection alone, only one patient with a 6 mm type-3 G1 gNEN developed lymph node and liver metastases during a median follow-up of 32 months. For this reason, the authors concluded that gastrectomy with lymph node dissection should still be considered as the treatment of choice for type-3 gNENs; however, endoscopic resection may represent an option for type-3 G1 gNENs ≤ 10 mm in size, if confined to the submucosa [57]. According to the latest ENETS guidelines [8], endoscopic resection is possible in cases of type-3 G1 gNENs under 10 mm, on the condition that previously performed anatomical (MRI, CT scan, EUS) and functional (68GalliumPET-CT scan in case of low-G NENs, FDG PET-CT scan in case of higher-G NENs) imaging studies exclude the presence of distant or lymph node metastasis, as well as muscular invasion. Endoscopic treatments, such as EMR or ESD techniques, can be also proposed for patients with high operatory risk or in selected cases of <15 mm low-G2 (Ki-67 < 10%) type-3 gNENs with favorable histology (no lymphovascular nor muscularis propria invasion) [58]. In the case of R1 endoscopic resection of a type-3 gNEN, surgery and, in particular, gastric wedge resection represents the most appropriate rescue therapy. After endoscopic resection, a strict follow-up with thoracoabdominal CT and liver MRI must be planned, in order to detect the eventual recurrence of the tumor.

### 3.3. Surgery

The main risk factors for gNENs’ aggressive behavior are lesion size, depth of invasion, proliferation index or mitotic count, and the presence of distant metastases [59].

For these reasons, according to the latest European guidelines [8,26], type-1 and type-2 gNENs larger than 10 mm or with a higher proliferation index (G2 lesion with ki 67 >10%) should undergo staging by EUS. In case of evidence at EUS of lymph node metastases or invasion of the muscularis propria, it is mandatory to exclude the presence of distant metastases by axial anatomic and functional imaging techniques (CT-scan/MRI and 68GalliumPET-CT scan/PDG-PET-CT scan). This extensive staging should be performed upfront in cases of type-1 or type-2 G3 lesions and/or in cases of large lesions, conventionally those >20 mm [60].

In cases where distant metastasis and vascular macroinvasion are excluded, surgery should be considered; it finds its indication for lesions in which endoscopic resection would hardly guarantee R0. Specifically, the surgical approach is recommended in tumors >20 mm or with suspected muscolaris propria invasion (either on axial imaging or EUS). In addition, it should be considered for high-proliferation-index tumors (G3 gNENs, with Ki 67 > 20%) [8]. In cases of the absence of lymph node metastasis at pre-operative staging, limited resection with local nodal sampling is the preferred surgical strategy, while total gastrectomy associated with D2 lymphadenectomy is the best choice if the patients have documented lymph node metastases.

Furthermore, in the case of R1 endoscopic incomplete resection of localized tumors, alongside with a step-up approach that includes progressively more radical endoscopic techniques (EMR, ESD, and EFTR), surgery must be considered [60]. 

Finally, surgery represents the therapeutic option of choice in case of the need for radicalization of the excision of the lesion if the final pathology of the resected specimen (regardless of the technique used for the first resection, whether endoscopic or surgical) demonstrates characteristics of greater aggressiveness than the preoperative staging (lymphovascular invasion, high Ki 67 index). If lymphovascular invasion is highlighted on a surgical specimen deriving from a limited resection with local nodal sampling, total gastrectomy associated with D2 lymphadenectomy should be preferred as the radicalization technique. Also, in these scenarios, it is critical to exclude distant metastatic disease by complete restaging [8,26,61].

Once again, it is useful to highlight that the therapeutic efficacy of the excisional treatment of type-2 gNENs, either endoscopic or surgical, greatly depends on the management of the underlying ZES and, possibly, MEN1 syndrome [23,24].

Given their more significant risk of metastasis [7], type-3 gNENs are traditionally addressed with a surgical approach [58]. Regardless of their size and their proliferation index upon biopsy-deriving pathology assessment, any type-3 gNEN needs to undergo complete anatomical and functional staging with a CT-scan/MRI and a 68GalliumPET-CT scan/PDG-PET-CT scan to identify the best treatment approach [8,59].

If preoperative staging does not reveal distant metastases or macroangioinvasion, surgery should be considered for the treatment of type-3 gNENs. In particular, in the absence of lymph node metastases and infiltration of the muscularis propria, <20 mm type-3 gNENs with a low proliferative index (Ki 67 < 3%, G1) are indicated for gastric wedge resection without standard lymphadenectomy [62]. These indications are currently much debated, and there is no unanimity in defining the optimal cutoffs regarding both the tumor size and the proliferative index of the lesion. The possibility that the parenchymal sparing surgical technique could also be proposed for selected cases of G2 type-3 gNENs, or for laterally spreading non-infiltrating lesions, is not excluded [56,62].

A rare exception to these indications is represented by small <10 mm type-3 G1 gNENs, which can be evaluated case-by-case for endoscopic resection (usually an ESD) [55,57,63].

A more radical approach adopting total or subtotal gastrectomy with D2 lymphadenectomy, on the other hand, represents the procedure of choice as a first-line therapy for patients with localized G2-G3 type-3 gNENs (Ki67 > 20%) and/or tumor sizes over 20 mm; furthermore, this radical surgical resection technique should be preferred to wedge resection in all cases of type-3 gNENs with lymph node involvement at pre-operative staging [8,57].

As described for type-1 and type-2 gNENs, for type-3 gNENs, a step-up approach should also be strictly followed in case of greater aggressiveness of the lesion on the endoscopic or operative specimen, or in case of R1 resection margins [8].

### 3.4. Systemic Therapy

In cases of metastatic disease; evidence of recurrent disease despite multiple attempts of resection, either endoscopic or surgical; inability to guarantee an R0 resection; or large tumors, systemic therapy must be taken into account, even if data evaluating this approach are lacking [8]. In addition, the pharmacological therapeutic option should also be considered in the case of patients unfit for surgery/endoscopy.

Type-1 and type-2 gNENs are usually well-differentiated tumors, and for this reason, they express somatostatin receptors [6]. Data regarding therapy with somatostatin analogs (SSAs) for type-1 gNENs requiring systemic therapy showed a high complete response rate (25–100%) [64]. The most commonly used drugs are octreotide LAR (long-acting release) and lanreotide, at the usual posology of 30 mg/3–4 w and 60 mg/3–4 w, respectively. The minimum period of therapy has not yet been established with clarity, but relapses have frequently been observed after discontinuation of therapy [64]. Most experts believe that a response evaluation should be performed after at least 12 months of therapy [65]. Specifically, in the cases of type-2 gNENs sustained by irresectable gastrinomas, SSAs represent, together with PPI, the therapeutic cornerstone of the disease [66,67].

Alternative therapeutic options have been proposed for advanced or metastatic unresectable NENs: everolimus, an mTOR pathway inhibitor [68]; sunitinib, a multitargeted tyrosine kinase inhibitor (TKI) [69]; and other multikinase inhibitors such as Cabozantinib and Sorafenib, which are, at the moment, being tested in phase II clinical trials (NCT05048901 NCT00131911). Nevertheless, to the best of our knowledge, no specific evidence on gastric neoplasms is available to date.

Finally, the direct inhibition of gastrin receptors has been investigated in a phase II clinical trial in patients with hypergastrinemic neuroendocrine neoplasms of type 1 or 2. The gastrin receptor inhibitor netazepide resulted in a complete response in 30% of patients. Tumor relapse was observed in all patients after discontinuation of the drug [70]. Specific head-to-head trials comparing netazepide and SSAs are promising.

### 3.5. Other Therapeutic Options for Advanced, Metastatic, or Progressive gNENs

Cytotoxic chemotherapy can be chosen as a rescue therapy for metastatic, progressive type-1 and type-2 gNENs, or those that do not respond to other regimens. In most cases, type-1 and type-2 gNENs showing aggressive behavior are G3. In these cases, the commonly used cytotoxic drugs, either alone or in combination, are antimetabolites (capecitabine and 5-fluorouracil), alkylating agents (temozolomide), and anthracyclines (doxorubicin and epirubicin) [71].

In addition, well-differentiated neoplasms, such as type-1 and type-2 G1 or G2 gNENs, usually express somatostatin receptors; thus, peptide receptor radionuclide therapy (PRRT) represents an effective therapeutic option. It is a cyclical scheduled therapy based on radiopharmaceuticals created by conjugating a β-emitting radioisotope to an SSA; the drug is then administered to the patients and captured by the tumor [72]. Overexpression of somatostatin receptors is, thus, mandatory in order to guarantee the efficacy of this treatment. The most often used SSA radio-conjugates are 90Y-DOTATOC, 177Lu-DOTATATE, and 111In-DTPA.

Finally, locoregional treatment techniques for liver lesions have also been proposed for neuroendocrine metastases [73]. These include liver thermal ablation through radio frequencies, laser, or cryoablation [74], as well as intra-arterial therapies (IATs) such as TAE (trans-arterial embolization) [75], TACE (trans-arterial chemoembolization) [76], and TARE (trans-arterial radioembolization) [77]. Most of the evidence available to date, however, refers to advanced or metastatic neuroendocrine neoplasms of pancreatic origin [78]. New cohort studies on patients with gNENs and, possibly, head-to-head trials comparing various therapeutic options for advanced disease should be conducted.

## 4. Proposed Practical Management of Well-Differentiated gNENs

Based on the currently available data and our clinical experience, we have formulated an algorithm for the endoscopic management of localized gNENs (Figure 3 and Figure 4).

### 4.1. Type-1 and Type-2 gNENs

For type-1 and type-2 gNENs, the endoscopic approach is recommended in cases of isolated lesions (less than 6 visible lesions) that have a low Ki67 proliferation index (G1/G2). In the presence of more than six lesions or recurrent disease, a systemic approach with SSAs should be considered [64].

Small G1 gNENs less than 5 mm in diameter, and, probably G2 gNENs with Ki67 <10%, can be actively observed endoscopically. Lesions between 5 and 10 mm can be treated with endoscopic removal upfront, without EUS staging, because of their low risk of submucosal/muscular involvement. Taking into consideration the high number of recurrences reported in the literature, excisional biopsy and simple snare polypectomy should not been performed with therapeutic intent [34].

GNENs larger than 10–15 mm and/or non-low G2 lesions, either type-1 or type-2, are worthy of consideration for removal, and they should undergo EUS staging to determine the depth of tumor infiltration and to assess loco-regional lymph node involvement. G3 and/or >20 mm lesions should undergo complete staging through EUS, CT-scan/MRI, and PET-CT scan to identify the most appropriate therapeutic strategy. These aggressive neoplasms represent a very small proportion of all gNENs.

For endoscopic resection in type-1 or type-2 gNENs, with a median diameter below 10/15 mm and absence of submucosal involvement, both EMR and ESD represent suitable treatment methods. Overall, EMR (including modified EMR) presents the advantages of a higher safety profile and a shorter procedure time, with a possible lower rate of complete resection. On the other hand, ESD is a technical, demanding procedure with a longer procedural time and a higher rate of R0 resection, especially in the presence of submucosal invasion, and possibly a lower risk of recurrence.

In addition, modified EMR (cap band assisted and underwater EMR) shows a promising rate of complete resection in small series, and its application should be encouraged.

In cases of lesions above 15 mm, evidence of submucosal infiltration, risk factors (G2, ulcerated lesion), or signs of fibrosis, ESD represents the treatment of choice. Even though there is no solid evidence, EFTR should be considered a viable option in cases of well-differentiated lesions with low proliferation indices (G1), but with evidence of invasion of the superficial layers of the muscularis propria, which would have a surgical indication. However, EFTR’s effect on gNENs still needs to be deeply investigated.

### 4.2. Type-3 gNENs

For type-3 gNENs, given their elevated risk of nodal and distant metastasis, a more aggressive approach and careful patient selection are advisable. While surgery traditionally remains the preferred therapeutic strategy, endoscopic resection is feasible for type-3 G1 gNENs under 10 mm, and, exceptionally, under 15 mm. This is contingent upon prior anatomical (MRI, CT scan, EUS) and functional (68GalliumPET-CT scan and/or FDG PET-CT scan) imaging studies confirming the absence of distant and lymph node metastasis and muscular invasion. For lesions between 5 and 10 mm, modified EMR and ESD are potential options. For lesions with sizes between 10 and 15 mm, ESD is the preferred choice, with EFTR potentially playing a role.

## 5. Follow-Up

All currently available guidelines recommend a careful follow-up for gNEN. However, a grey area remains regarding the precise timing and modalities of follow-up for these neoplasms. Establishing an appropriate observation plan requires clinicians to consider various factors, including tumor characteristics (such as gNEN type, size, and grading), patient characteristics (age, comorbidities, and attitude on treatments and follow-up), and treatment-related factors such as type of treatment, radicality, and risk of recurrence.

### 5.1. Type-1 and Type-2 gNENs

Patients with type-1 and type-2 lesions undergoing active observation should have an upper endoscopy performed every 6 months for the first year after diagnosis, and then 1–2 years in presence of stable disease. Reassessment for endoscopic treatment or systemic treatment should be considered in the case of gNEN progression regarding the size or number of lesions [8].

For patients who have undergone endoscopic R0 resection of a type-1 or type-2 gNEN, it is recommended to have an endoscopic follow-up every 12 months. A shorter interval could be advisable in the presence of risk factors such as incomplete resection, grading G2, or a size of >20 mm. Conversely, an extended interval can be considered after prolonged observation without evidence of relapse [8].

In general, no cross-sectional imaging is required for follow-up for small type 1 and type-2 gNENs [8].

### 5.2. Type-3 gNENs

The follow-up of patients who underwent surgical resection for type-3 gNENs is based on cross-sectional imaging (thoracoabdominal CT/liver-MRI), upper endoscopy, and/or EUS and functional imaging (68GalliumPET-CT scan and/or FDG-PET, depending on the tumor grade). When a total gastrectomy with lymphadenectomy is performed, the follow-up schedule adopted for gastric adenocarcinoma should be applied [62].

In patients undergoing endoscopic or surgical local resection, upper gastrointestinal endoscopy should be performed at 3 months to check the resection site. If this shows no macroscopic residual tumor, they should have regular follow-up visits with cross-sectional imaging and endoscopy/EUS. The frequency and choice of test will be influenced by the final tumor size and grade fitness, and in most cases, it will be possible to reduce the frequency of follow-up visits as time progresses after resection [8].

A 68GalliumPET-CT and/or FDG-PET scan, depending on the tumor grade, as well as biopsies, should be performed in the presence of a suspected disease relapse, but they are not routinely part of the follow-up program [8].

## 6. Discussion

The dynamic landscape of gNEN management is marked by evolving strategies and presents several management challenges and complexities.

The first step in navigating this complex terrain is distinguishing the types of gNENs, as this represents the starting point for therapeutic decisions. Gastric neuroendocrine neoplasms encompass a spectrum of highly heterogeneous diseases with vastly different prognoses. Their biological behavior varies from the indolent and non-aggressive type-1 gNENs, where there is a risk of overtreatment, to the more aggressive type-3 and G3 gNENs, which can potentially be metastatic tumors. In the latter cases, the risk lies in underestimating the prognosis, potentially leading to undertreatment.

While less aggressive approaches, such as active follow-up and minimally invasive endoscopic therapies, are gaining prominence, particularly in the management of type-1 gNENs, for patients requiring more invasive approaches, defining specific criteria for patient selection, refining endoscopic techniques, and exploring emerging technologies like EFTR require comprehensive investigation.

Moreover, even if a minimally invasive approach for type 1 gNENs is feasible through endoscopic management, recurrence rates after endoscopic removal of gNENs remain a major challenge. Although these recurrences are common, their clinical significance and impact on overall patient survival are uncertain. This raises critical questions regarding the appropriate response to recurrences and emphasizes the need for long-term, prospective studies to assess their true clinical relevance.

On the other hand, for more aggressive diseases such as type 3 gNENs, several challenges may arise. Identifying negative prognostic factors, including Ki67 proliferation index, tumor size, and depth of invasion, is crucial in order to assess the malignant potential of these tumors accurately. This helps in reserving aggressive treatments and strategies for larger, type-3 gNENs.

For these reasons, collaboration among clinicians, researchers, and patients is critical to creating a learning environment that optimizes patient care and is consistent with the evolving landscape of gNEN management.

## 7. Conclusions

In conclusion, the management of gastric neuroendocrine neoplasms presents intricate challenges, requiring tailored approaches based on tumor characteristics. The non-operative approach for small type-1 gNENs seems to be feasible and should be promoted, even if further studies are needed to validate its efficacy. However, even when endoscopically resected, gNENs often exhibit a high recurrence rate after endoscopic removal. Nevertheless, the impact of recurrence remains uncertain, with emerging evidence suggesting a minimal impact on overall mortality.

On the other hand, it is important to recognize the negative prognostic factors and features in order to identify those rare cases requiring more aggressive approaches or systemic therapies, and not to underestimate the importance of the rare cases of aggressive disease.

Therefore, looking ahead, the evolution towards less aggressive approaches, maintaining an active follow-up process, and adopting minimally invasive endoscopic therapies might replace surgery, particularly for type-1 gNENs, representing a potential future perspective.

This dynamic landscape underscores the need for ongoing research to refine therapeutic strategies for these complex neoplasms.

## Figures and Tables

**Figure 1 medicina-59-01757-f001:**
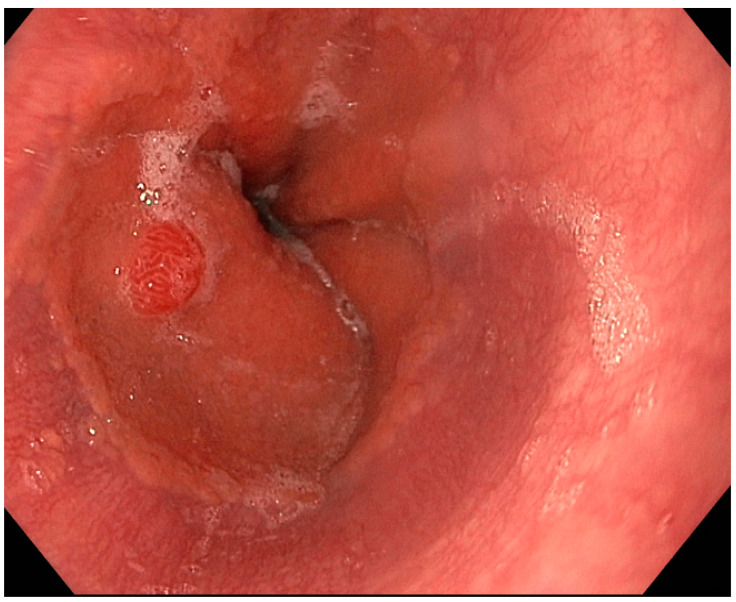
Endoscopic aspect of a type-1 gastric neuroendocrine neoplasm.

**Figure 2 medicina-59-01757-f002:**
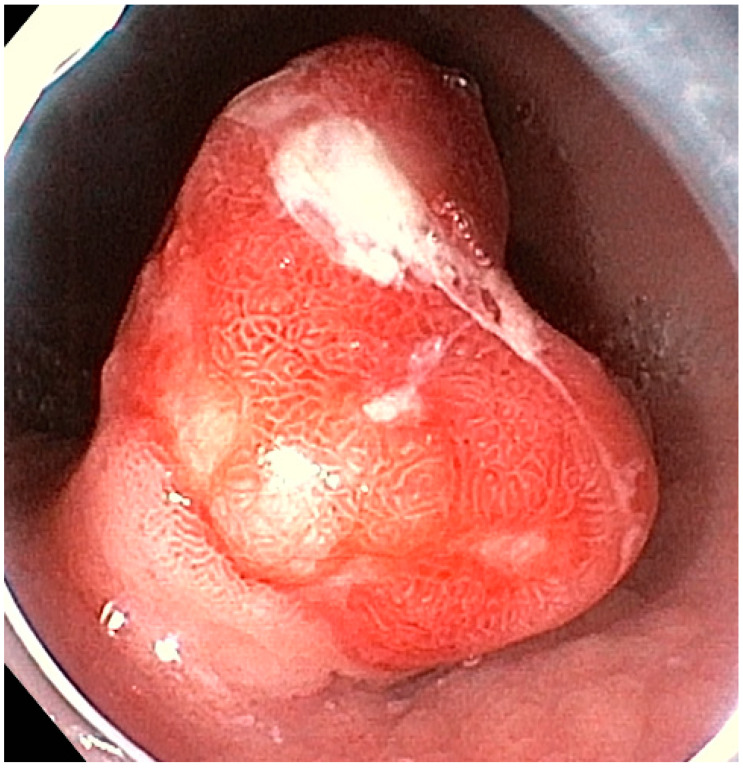
Endoscopic aspect of a type-3 gastric neuroendocrine neoplasm.

**Figure 3 medicina-59-01757-f003:**
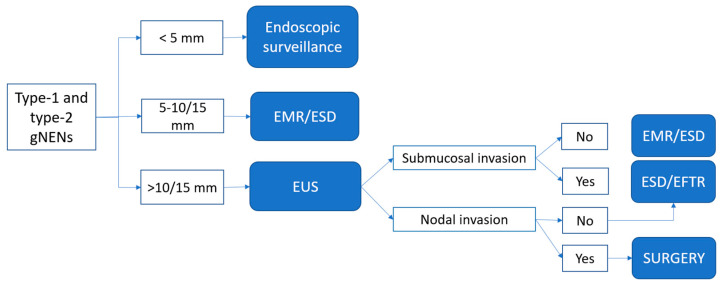
Proposed algorithm for localized, isolated type-1 and type-2 gNEN treatment.

**Figure 4 medicina-59-01757-f004:**
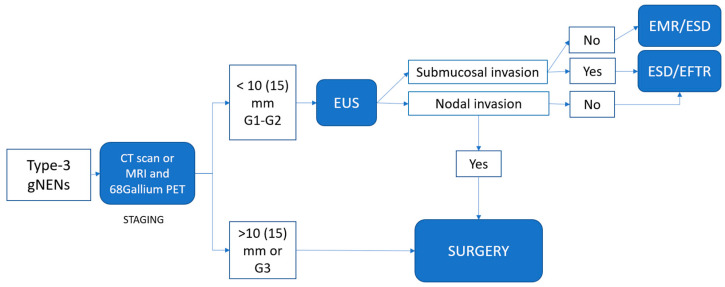
Proposed algorithm for localized, isolated type-3 gNEN treatment.

## Data Availability

Data sharing is not applicable.

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
