# Peer review of "Therapeutic Challenges for Gastric Neuroendocrine Neoplasms: Take It or Leave It?"

_medicina, 2023, doi:10.3390/medicina59101757_

Round 1

Reviewer 1 Report

The authors present a comprehensive review regarding the therapeutic options for gastric neuroendocrine neoplasms. Though rare encountered in clinical practice, those tumors present specific challenges and peculiarities of diagnostic and treatment. The references are up-to date, and the authors propose an algorithm that might help the surgeon in clinical practice.

Some minor issues:

1. Figure 1 and 2 should be inserted in the text, where apropiate, not presented as separate section.

2. A paragraph with discussions should be included before Conclusions

Author Response

Thank you for your valuable feedback on the article on therapeutic options for gastric neuroendocrine neoplasms. Your insights are greatly appreciated and can certainly improve the overall quality of the manuscript.

Regarding the minor issues risen:

Reviewer Comment 1: Figure 1 and 2 should be inserted in the text, where appropriate, not presented as a separate section.

Response: Thank you for your suggestion. We appreciate your advice to integrate Figure 1 and 2 directly into the text at relevant points in the manuscript. We made the necessary revisions to incorporate these figures appropriately within the text to enhance the flow of the article.

Reviewer Comment 2: A paragraph with discussions should be included before Conclusions.

Response: We value your input, and we agree that including a dedicated discussion section before the conclusions would enhance the structure of the article. We made the appropriate modifications to introduce a discussion section.

Reviewer 2 Report

Review of the paper “Therapeutic Challenges for Gastric Neuroendocrine Neoplasms: Take it or Leave it?”

The article proposed by Cavalcoli et al comprehensively presents, review and compare the available therapeutic strategies for gNENs.

Nowadays, there are several methods embedded and approved by several guidelines as well as national committees to use different endoscopic resective techniques (such as endoscopic mucosal dissection, full-thickness resection and mucosal resection) developed and with a high rate of complete resection for advanced and more aggressive gastric lesions.

I read the most recent reviews in the field, also cite by the author:

Exarchou, K.; Hu, H.; Stephens, N.A.; Moore, A.R.; Kelly, M.; Lamarca, A.; Mansoor, W.; Hubner, R.; McNamara, M.G.; Smart, 517 H.; et al. Endoscopic Surveillance Alone Is Feasible and Safe in Type I Gastric Neuroendocrine Neoplasms Less than 10 Mm in 518 Diameter. Endocrine 202278, 186–196, doi:10.1007/s12020-022-03143-3. 

Roberto, G.A.; Rodrigues, C.M.B.; Peixoto, R.D.; Younes, R.N. Gastric Neuroendocrine Tumor: A Practical Literature Review. 544 World J. Gastrointest. Oncol. 202012, 850–856, doi:10.4251/wjgo.v12.i8.850. 

I think that the author has provided a possible therapeutic algorithm for gNENs which is shown in figure 1.

There are no concerns in my opinion regarding the manuscript.

Author Response

We thank the reviewer for his valuable feedback on the article on therapeutic options for gastric neuroendocrine neoplasms, we are glad he appreciated our paper. 

Reviewer 3 Report

Authors aims to comprehensively review and compare the available management and therapeutic strategies for the treatment of gNENs, and at providing a possible therapeutic algorithm for gNENs in this study.

A very broad and comprehensive review article. I congratulate the authors for their efforts. I suggested some minor revisions. I wish success to the authors

Comments

1: You need to explain how you did the literature review in the methods section, a separate methods section needs to be made.
Sample paragraph:  ‘’Methods: Literature review using the following databases: Medline/PubMed, Cochrane Library …. Search terms were: gastric neuroendocrine tumor, treatment….. From the selected articles, X  were included in this review…..

2: If you have Gastric Neuroendocrine neoplasm endoscopic images, you can add them to the article according to GNET types.

3: ‘’Consequently, type-1 gNENs represent relevant 58 complication of AIG, with an annual cumulative risk of approximately 5.7% according to 59 recent data (Massironi S. et al. – Epub ahead of print).’’

Comment: Add reference number to line 58-60

4: Line 209-213 ‘’Recently Kim et al, in a 209 small study on subepithelial lesions including gastric and duodenal NENs, reported en 210 bloc resection rates and complete resection rates of around 100%. The median size of the 211 resected lesions was 9 mm, and the mean procedural time was 3.2 min. No adverse 212 events were observed and no recurrence at 3 month-follow-up endoscopy [47].’’

Comment: Add correct reference, not 47

5: Finally, how should GNETs be monitored after treatment? You can explain this information in a separate paragraph.

 Minor editing of English language required

Author Response

Thank you for your constructive criticism on the article, your insights can certainly improve the overall quality of the manuscript.

Regarding the concerns risen:

Reviewer Comment 1: You need to explain how you did the literature review in the methods section, a separate methods section needs to be made.
Sample paragraph:  ‘’Methods: Literature review using the following databases: Medline/PubMed, Cochrane Library …. Search terms were: gastric neuroendocrine tumor, treatment….. From the selected articles, X  were included in this review…..

Response: As suggested we added a “Material and Methods” section explaining how the literature review have been performed (Page 4; lines 100-117)

Reviewer Comment 2: If you have Gastric Neuroendocrine neoplasm endoscopic images, you can add them to the article according to GNET types.

Response: We thank the reviewer for his suggestion the required images have been added (see figure 1 and 2)

Reviewer Comment 3: ‘’Consequently, type-1 gNENs represent relevant 58 complication of AIG, with an annual cumulative risk of approximately 5.7% according to 59 recent data (Massironi S. et al. – Epub ahead of print).’’ Comment: Add reference number to line 58-60

Response: The pertinent reference has been provided (see reference 30).

Reviewer Comment 4: Line 209-213 ‘’Recently Kim et al, in a 209 small study on subepithelial lesions including gastric and duodenal NENs, reported en 210 bloc resection rates and complete resection rates of around 100%. The median size of the 211 resected lesions was 9 mm, and the mean procedural time was 3.2 min. No adverse 212 events were observed and no recurrence at 3 month-follow-up endoscopy [47].’’Comment: Add correct reference, not 47

Response: We thank the reviewer for having highlighted this inconsistency the reference has been corrected.

Reviewer Comment 5: Finally, how should GNETs be monitored after treatment? You can explain this information in a separate paragraph.

Response: thank you for your helpful suggestion, a paragraph on gNENs follow-up has been added (Pages 13-14, Lines 509-544).